# Integrated Transcriptomic and Proteomic Study of the Mechanism of Action of the Novel Small-Molecule Positive Allosteric Modulator 1 in Targeting PAC1-R for the Treatment of D-Gal-Induced Aging Mice

**DOI:** 10.3390/ijms25073872

**Published:** 2024-03-30

**Authors:** Lili Liang, Shang Chen, Wanlin Su, Huahua Zhang, Rongjie Yu

**Affiliations:** 1Department of Cell Biology, College of Life Science and Technology, Jinan University, Guangzhou 510632, China; 2Department of Medical Genetics, Guangdong Medical University, Dongguan 523808, China; 3Guangdong Province Key Laboratory of Bioengineering Medicine, Guangzhou 510632, China; 4Guangdong Provincial Biotechnology Drug & Engineering Technology Research Center, Guangzhou 510632, China; 5National Engineering Research Center of Genetic Medicine, Guangzhou 510632, China

**Keywords:** pituitary adenylate cyclase-activating polypeptide (PACAP), pituitary adenylate cyclase-activating polypeptide receptor 1 (PAC1-R), anti-aging, neuroprotective effects, transcriptomic, proteomic

## Abstract

Small-molecule positive allosteric modulator 1 (SPAM1), which targets pituitary adenylate cyclase-activating polypeptide receptor 1 (PAC1-R), has been found to have a neuroprotective effect, and the underlying mechanism was explored in this study. First, using a D-galactose (D-gal)-induced aging mouse model, we confirmed that SPAM1 improves the structure of the hippocampal dentate gyrus and restores the number of neurons. Compared with D-gal model mice, SPAM1-treated mice showed up-regulated expression of Sirtuin 6 (SIRT6) and Lamin B1 and down-regulated expression of YinYang 1 (YY1) and p16. A similar tendency was observed in senescent RGC-5 cells induced by long-term culture, indicating that SPAM1 exhibits significant in vitro and in vivo anti-senescence activity in neurons. Then, using whole-transcriptome sequencing and proteomic analysis, we further explored the mechanism behind SPAM1’s neuroprotective effects and found that SPAM is involved in the longevity-regulating pathway. Finally, the up-regulation of neurofilament light and medium polypeptides indicated by the proteomics results was further confirmed by Western blotting. These results help to lay a pharmacological network foundation for the use of SPAM1 as a potent anti-aging therapeutic drug to combat neurodegeneration with anti-senescence, neuroprotective, and nerve regeneration activity.

## 1. Introduction

Pituitary adenylate cyclase-activating polypeptide receptor 1 (PAC1-R) is a G protein-coupled receptor (GPCR) that has a higher affinity for pituitary adenylate cyclase-activating polypeptide (PACAP) than for vasoactive intestinal peptide (VIP) [1]. PAC1-R is widely distributed in the central nervous system, and PAC1-R expression is particularly high in neurogenic regions of the adult brain, such as the subventricular zone of the olfactory bulb or the dentate gyrus of the hippocampus [2]. PACAP’s anti-apoptotic effects, induction of neural stem cell differentiation [3], and antioxidant effects are mediated by PAC1-R [4]. PACAP inhibits pathological processes in Alzheimer’s disease (AD) and Parkinson’s disease (PD) models through PAC1-R [5], and thus, PAC1-R could be a target for the development of drugs for neurodegenerative diseases.

In a previous study, we used computerized molecular docking to screen for the docking of doxycycline and minocycline with PAC1-R. Our experiments verified that both had neuroprotective effects and that doxycycline was a positive allosteric modulator targeting PAC1-R [6,7]. In addition, through computerized virtual screening and laboratory screening, we obtained new data on a positive allosteric modulator targeting PAC1-R named small positive allosteric modulator 1 (SPAM1) (patent No.: CN202210388027.7; molecular formula: C_20_H_19_N_3_O_4_). Additionally, SPAM1 targeted PAC1-R with significantly higher affinity than DOX, and SPAM1 had more effective cytoprotective activity [8]. We showed in our latest study that SPAM1 induces the nuclear translocation of PAC1-R, which exerts neuroprotective effects by regulating neuronal restriction silencing factor (NRSF) [9].

Sirtuin 6 (SIRT6) is a member of the Sirtuin family and promotes dendritic morphogenesis in hippocampal neurons during developmental stages [10]. SIRT6 is associated with the aging process, has DNA-repair [11] and neuroprotective effects, and exhibits severely reduced levels in patients with Alzheimer’s disease [12]. YinYang 1 (YY1) is a zinc finger protein that belongs to the GLI-Küppel gene family [13], and its expression increases with age [14]. Global knockout of YY1 resulted in mouse embryonic death, and mice exhibited motor deficits and cognitive deficits in a YY1 conditional knockout mouse model [15]. Studies have shown that a decline in LaminB1 levels during the aging process leads to senescence and neurodegenerative diseases such as Alzheimer’s and Parkinson’s disease [16]. Decreased LaminB1 expression causes an age-dependent decrease in hippocampal neural stem cell activity [17]. Both p16 and Lamin B1 are recognized as biomarkers of cellular senescence [18].

In the present study, we demonstrated that SPAM1 up-regulated the expression of SIRT6 and Lamin B1 and down-regulated the expression of YY1 and p16 in the hippocampus of aging D-gal-induced model mice. In this study, we combined whole-transcriptome sequencing and proteomic analysis to further investigate the mechanism through which SPAM1 exerts neuroprotective effects, thereby laying the pharmacological foundation for the use of SPAM1 as a therapeutic agent for neurodegenerative diseases.

## 2. Results

### 2.1. SPAM1 Prevents D-Gal-Mediated Reduction in the Number of Mouse Hippocampal Neurons

The dentate gyrus functions as a “gatekeeper” regulating the influx of information into the hippocampus. It plays a critical role in learning, memory consolidation, spatial navigation, and emotional regulation [19]. HE staining showed that the number of neurons in the hippocampus of the mouse brain was lower after treatment with D-gal compared with the untreated group. Nevertheless, in the group treated with SPAM1 in varying concentrations, the original number of neurons in the hippocampal region was restored, unlike in the D-gal group. The 0.1 μmol/kg/d and 100 μmol/kg/d SPAM1 groups showed significant effects (Figure 1), which suggests that treatment with SPAM1 reverses the D-gal-mediated reduction in the number of hippocampal neurons in the mouse brain.

### 2.2. SPAM1 Ameliorated RGC-5 Cell Senescence

To investigate the effect of SPAM1 on RGC-5 cell senescence, RGC-5 cells were cultured for 40 days, followed by incubation with SPAM1 (1–100 μM) for 24 h and an assay using β-gal staining. It was observed that the number of β-gal-positive RGC-5 cells was significantly reduced by SPAM1 (Figure 2), indicating that SPAM1 has inhibitory effects on senescent RGC-5 cells.

### 2.3. SPAM1 Treatment Up-Regulated the Expression of SIRT6 and Lamin B1 and Down-Regulated the Expression of YY1 and p16 in RGC-5 Cells and Mouse Brains

In this study, we found that, in RGC-5 cells, the expression of senescence-related proteins was affected by SPAM1, which down-regulated the expression of p16 and up-regulated the expression of Lamin B1 (Figure 3a–c). In the present study, we further investigated the mechanism of SPAM1 and found that it down-regulated the expression of YY1 and up-regulated the expression of SIRT6 (Figure 3d). Interestingly, we found that SPAM1 similarly down-regulated YY1 expression and up-regulated SIRT6 expression in non-senescent 10-day-old RGC-5 cells (Figure 3e). These results suggest that SPAM1 exerts an anti-senescence effect by regulating the expression of YY1 and SIRT6 in RGC-5 cells.

Immunohistochemical experiments showed that the expression of SIRT6 and Lamin B1 in the hippocampus of D-gal-treated mice was decreased compared with that in the NOR group, while the use of SPAM1 increased the expression of Lamin B1 (Figure 4a) and SIRT6 (Figure 4c). The expression of p16 and YY1 was higher in the hippocampus of D-gal-treated mice compared with the NOR group, and the use of SPAM1 decreased the expression of P16 (Figure 4b) and YY1 (Figure 4d). These results demonstrate the anti-aging and neuroprotective effects of SPAM1 at the brain-tissue level.

### 2.4. Analysis of Differentially Expressed mRNAs, lncRNAs, circRNAs, and miRNAs

The screening criteria were as follows: |Fold Change| ≥ 1.5 and *p* value < 0.05 for differentially expressed mRNAs, |Log2(Fold Change)| ≥ 1 and FDR < 0.05 for differentially expressed LncRNAs and circRNAs, and |Log2(Fold Change)| ≥ 1 and *p* value < 0.05 for differentially expressed miRNAs.

Three samples each were randomly selected from the D-gal group (subsequently named the DA group) and the 100 µmol/kg/day SPAM1 group (subsequently named SP group). The whole-transcriptome sequencing results showed that there were a total of 252 differentially expressed RNAs in the SP group, which was higher than in the DA group: 107 differentially expressed mRNAs, including 45 up-regulated and 62 down-regulated mRNAs; 44 differentially expressed lncRNAs, including 20 up-regulated and 24 down-regulated lncRNAs; 85 differentially expressed circRNAs, including 42 up-regulated and 43 down-regulated lncRNAs; 16 differentially expressed miRNAs, including 7 up-regulated and 9 down-regulated miRNAs; and 43 down-regulated lncRNAs. The differentially expressed miRNAs totaled 16, including 7 up-regulated and 9 down-regulated miRNAs. Hierarchical cluster analysis aims to cluster RNAs with identical or similar expression patterns. The heatmap consistently clustered the three samples from the DA and SP groups, indicating that differentially expressed mRNAs, lncRNAs, circRNAs, and miRNAs are highly reproducible. These results confirm the reliability of the transcriptome analysis in this study (Figure 5a–d). The volcano plot permits the observation of distinctions in the expression of mRNAs, lncRNAs, circRNAs, and miRNAs between two groups, as well as the statistical significance of the differences (Figure 6a–d).

### 2.5. Functional Annotation and Enrichment Analysis of Differentially Expressed mRNAs

A total of 107 differentially expressed mRNAs were identified in this study, based on which GO and KEGG enrichment analyses were performed. The numbers of differentially expressed mRNAs annotated in the GO and KEGG databases were 75 and 19, respectively (Figure 7a). This study determined the 20 pathways with the least significant Q values in the KEGG pathway enrichment analysis of differentially expressed mRNAs, which were enriched in the renin–angiotensin system, phospholipase D signaling pathway, neuroactive ligand−receptor interaction, etc. (Figure 8a).

### 2.6. Functional Annotation and Enrichment Analysis of Differentially Expressed LncRNA Target Genes

Functional annotation showed that a total of 492 significantly differentially expressed LncRNA target genes were annotated, with 442 (Figure 7b) and 87 annotated in the GO database and KEGG databases, respectively. This study determined the 20 pathways with the least significant Q values in the KEGG pathway enrichment analysis of differentially expressed LncRNA target genes, which were enriched in the olfactory transduction pathway, longevity-regulating pathway—multiple species, longevity-regulating pathway, circadian entrainment pathway, cell cycle pathway, etc. (Figure 8b).

### 2.7. Functional Annotation and Enrichment Analysis of Host Genes That Differentially Express circRNAs

Functional annotation showed that a total of 71 (Figure 7c) and 30 significant host genes that differentially expressed circRNAs were annotated in the GO and KEGG databases, respectively. This study determined the 20 pathways with the least significant Q values in the KEGG pathway enrichment analysis of host genes that differentially express circRNAs, which were enriched in ubiquitin-mediated proteolysis, the synaptic vesicle cycle, the hedgehog signaling pathway, circadian rhythm, the cGMP-PKG signaling pathway, the cAMP signaling pathway, etc. (Figure 8c).

### 2.8. Functional Annotation and Enrichment Analysis of Differentially Expressed miRNA Target Genes

Functional annotation showed that a total of 4493 differentially expressed miRNA target genes were annotated, with 4246 (Figure 7d) and 975 annotated in the GO and KEGG databases, respectively. This study determined the 20 pathways with the least significant Q values in the KEGG pathway enrichment analysis of differentially expressed miRNA target genes, which were enriched in protein digestion and absorption, lysine degradation, hematopoietic cell lineage, the cAMP signaling pathway, the calcium signaling pathway, basal cell carcinoma, etc. (Figure 8d).

### 2.9. Whole-Transcriptome Association Analysis

The GO enrichment analysis showed that differentially expressed mRNAs, LncRNA target genes, circRNA host genes, and miRNA target genes are all involved in synapses, synapse parts, responses to stimuli, developmental processes, signaling, cell proliferation, etc. Both differentially expressed LncRNA target genes and miRNA target genes are involved in antioxidant activity (Figure 7a–d).

PageRank, a classical algorithm in random walk, was used to obtain the scores (importance) of all nodes in the network. The points in the network with scores ranked in the top 0.1, that is, the key RNAs, were selected as the focus of this study. In this study, genes in the key RNAs were analyzed for pathway enrichment, and the five most significantly enriched pathways were selected. The gene-to-gene relationships in these five pathways were extracted and integrated into a pathway network (Figure 9a).

CircRNAs are mainly produced by pre-mRNAs through variable shear processing, and the gene that produces circRNAs is the host gene. miRNAs control gene expression by silencing and degrading their target gene molecules. In this study, targeting relationship combined analysis was performed for differentially expressed RNAs.

In this study, differentially expressed circRNAs represented the center of the joint analysis of targeting relationships; the intersection and host gene of the differentially expressed circRNAs and the differentially expressed miRNAs with which the differentially expressed circRNAs had a targeting relationship are presented in the form of a Venn diagram. The intersection of the first and third terms is 65 circRNAs, and the intersection of the second and third terms is 18 circRNAs (Figure 9b).

Next, we performed a targeting relationship association analysis centered on differentially expressed miRNAs. The intersection of differentially expressed miRNAs, all miRNAs targeted by differentially expressed mRNAs, and all miRNAs targeted by differentially expressed circRNAs is presented as a Venn diagram. The intersection of the first and third items is 6 miRNAs, of the second and third items is 890 miRNAs, and of the three items is 7 miRNAs (Figure 9c).

Finally, the present study is centered on differentially expressed mRNAs for the joint analysis of targeting relationships. The intersection of differentially expressed mRNAs, all target genes of differentially expressed miRNAs, and all host genes of differentially expressed circRNAs is presented as a Venn diagram. The intersection of the second and third terms is 26 mRNAs (Figure 9d).

### 2.10. Analysis of Differentially Expressed Proteins

In this study, further proteomic analysis of the DA and SP groups was performed to look for differentially expressed proteins in mouse brain tissues after treatment with SPAM1. A fold change of ≥1.2 or ≤0.83 with a *p*-value of <0.05 was used as a screening criterion. Compared with the DA group, there were 41 differentially expressed proteins in the SP group, of which 32 were up-regulated and 9 were down-regulated (Figure 10a,b). Proteins that were up-regulated included barrier-to-autointegration factor, calcium/calmodulin-dependent protein kinase type II subunit alpha, calcium/calmodulin-dependent protein kinase type II subunit beta, neurofilament light polypeptide (NEFL), neurofilament medium polypeptide (NEFM), neurofilament heavy polypeptide, peripherin, apolipoprotein D, and leucine-rich repeat-containing protein 4.

To further assess the connections between differentially expressed proteins, this study also mapped the differentially expressed protein string network (Figure 10c).

### 2.11. Functional Annotation and Enrichment Analysis of Differentially Expressed Proteins

In this study, we functionally annotated 41 differentially expressed proteins (39 and 41 in the GO and KEGG databases, respectively). The GO enrichment analysis showed that the differentially expressed proteins were involved in synapses, synapse parts, responses to stimuli, developmental processes, signaling, cell proliferation, etc. (Figure 11a). This study determined the 20 pathways with the least significant Q values in the KEGG pathway enrichment analysis of differentially expressed proteins, which were enriched in the neurotrophin signaling pathway, circadian entrainment, long-term potentiation, olfactory transduction, the GnRH signaling pathway, etc. (Figure 11b).

### 2.12. Combined Whole-Transcriptome and Proteome Analysis

In this study, the Spearman correlation coefficients were calculated based on the eigenvector values of the protein modules and differential gene expression, and the interactions between differential genes and differentially expressed proteins were determined using Spearman correlation analysis to further explore the mechanism of action between strongly correlated genes and proteins. In this study, the 50 genes with the smallest differential gene FDR values in the transcriptomic analysis and the 50 proteins with the smallest differential protein *p*-values in the proteomic analysis were screened for the next correlation analysis (Figure 12a).

This study also screened 15 KEGG pathways, including the calcium signaling pathway, neuroactive ligand–receptor interactions, and cholinergic-like synapses, which were common in the transcriptomics and proteomic analyses.

In this study, the 200 genes with the smallest differential gene FDR values in the transcriptomic analysis and the 50 proteins with the smallest differential protein *p*-values in the proteomic analysis were screened for network regulation analysis. Based on the top 200 differential genes and top 50 differential proteins screened, the correlation results with absolute correlation values of >0.9 and *p*-values < 0.001 were further screened for network regulation mapping (Figure 12b).

### 2.13. Validation of Differentially Expressed Proteins

Based on the results of proteomic analysis, we further validated the protein expression after SPAM1 treatment in D-gal model mice. Consistent with the results of the proteomic analysis, the protein levels of neurofilament light polypeptide (NEFL) and neurofilament medium polypeptide (NEFM) were significantly up-regulated by SPAM1 (Figure 13).

## 3. Discussion

GPCRs can regulate gene transcription through an unconventional mode of signal transduction, whereby the GPCR cleaves and relies on its carboxyl-terminal structural domain to translocate to the nucleus [20]. In a previous study, we demonstrated that PAC1-R also has such a transduction pattern [9,21,22]. PAC1-R has a high affinity for PACAP due to a specific region of its N-terminal extracellular domain 1 (EC1) [1]. PAC1-R has become a target for drugs because of its crucial role in the nervous system [23]. In a previous study, we targeted PAC1-R-EC1 to screen doxycycline, minocycline, and SPAM1, and subsequent experiments demonstrated that SPAM1 exhibited more significant neuroprotective activity [6,7,8]; nevertheless, little is known about the mechanism of action of SPAM1.

D-gal was used to induce simulated aging in mice [24]. DG plays a key role in hippocampal memory formation, and DG lesions impair many hippocampus-dependent memory functions [25]. The accumulation of p16 occurs in senescent cells and the down-regulation of Lamin BI is another important feature of cellular senescence [18]. SIRT6 is implicated in DNA repair, and SIRT6-knockdown mice show a degenerative phenomenon similar to aging [26]. New findings also suggest that the changes observed in SIRT6-deficient brains also occur in the aging human brain, particularly in patients with Alzheimer’s, Parkinson’s, and Huntington’s diseases and amyotrophic lateral sclerosis. SIRT6 is a key regulator of mitochondrial function in the brain and interacts with transcription factor YY1 to influence mitochondrial function [27]. It is accompanied by a decrease in SIRT6 levels and an increase in YY1 levels in aged mice [14]. Data from clinical studies indicate that plasma levels of YY1 are significantly elevated in patients with major depressive disorder [28]. Notably, in a previous study, using chromatin immunoprecipitation, we showed that YY1 may be recruited by nuclear-translocated PAC1-R [21]. In this study, we showed that compared with mice in the NOR group, the number of D-gal-induced DG neurons in the hippocampus of the brain was reduced, and structural alterations were observed in the D-gal-induced senescent mouse model; these were accompanied by down-regulation of the expression of SIRT6 and Lamin B1 in DG, as well as up-regulation of the expression of YY1 and p16. However, mice treated with different concentrations of SPAM1 showed an increase in the number of DG neurons, structural recovery, and reversal of the expression of the four proteins in the appeal, that is, the up-regulation of SIRT6 and Lamin B1 and down-regulation of YY1 and p16. This suggests that SPAM1 plays a role in maintaining the structure of the hippocampus and protein expression; in other words, our results further support the theory that SPAM1 has neuroprotective effects and can be used to treat neurodegenerative disorders. Compared with normal RGC-5 cells cultured for only 10 days, senescent RGC-5 cells cultured for 40 days were accompanied by decreased expression of Lamin B1 and increased expression of p16. However, after using SPAM1, the expression of Lamin B1 was increased and that of p16 was decreased. We also carried out an exploration of its mechanism and found that SPAM1 down-regulated the expression of YY1 and up-regulated the expression of SIRT6. Notably, such results were also obtained in unsenescent 10-day-old RGC-5 cells, suggesting that this mechanism is also useful in normal neuronal cells. These results suggest that SPAM1 can exert anti-senescent effects not only in senescent neuronal cells, but also in normal neuronal cells; in corroboration with this, our transcriptomic analysis found that the KEGG pathway involved in SPAM1 is the longevity-regulating pathway. These results predict that SPAM1 has potential to be developed as an anti-aging drug.

The KEGG pathway of differentially expressed RNAs has also provided many new insights into the mechanism of SPAM1 action. For example, the brain renin–angiotensin system has been implicated in Alzheimer’s disease neuropathology [29], and it may serve as a novel potential therapeutic target for Alzheimer’s disease [30]. It has been shown that the phospholipase D signaling pathway is a key signaling pathway in Alzheimer’s disease [31]; additionally, PACAP activates PAC1, and PAC1 couples with the phospholipase D signaling pathway to trigger downstream effects [32]. Differentially expressed mRNAs were enriched to induce neuroactive ligand–receptor interaction in addition to the two pathways described above. It has also been shown that PACAP has an important role in the olfactory system, acting through PAC1-R [33]. PACAP is also involved in the regulation of circadian entrainment to light [34,35]. Differentially expressed LncRNA target genes were enriched in the above two pathways, in addition to the longevity-regulating pathway—multiple species, longevity-regulating pathway cell cycle pathway, etc. Interestingly, SIRT6 is also involved in the regulation of longevity [36]. This predicts that SPAM1 may have the potential to become a potent anti-aging drug. The ubiquitin–proteasome system is involved in regulating synaptic plasticity and memory formation [37], and its dysfunction has been linked to Alzheimer’s disease and dementia [38]. The PACAP/PKA pathway plays a role in tumor therapy by regulating the hedgehog signaling pathway [39,40]. In addition to these two pathways, host genes that differentially express circRNAs were also enriched in the synaptic vesicle cycle, cGMP-PKG signaling pathway, cAMP signaling pathway, etc., and in the circadian rhythm, as are differentially expressed LncRNA target genes. Similarly, differentially expressed miRNA target genes were enriched in protein digestion and absorption and the cAMP signaling pathway, in addition to the calcium signaling pathway. These transcriptomic analysis results indicate that SPAM1 exerts multiple positive effects, including anti-senescence, neuroprotective, and nerve regeneration, against neurodegeneration.

As for proteome sequencing, it was shown that, after SPAM1 treatment, barrier-to-autointegration factor, calcium/calmodulin-dependent protein kinase type II subunit alpha, calcium/calmodulin-dependent protein kinase type II subunit beta, neurofilament light polypeptide, neurofilament medium polypeptide, neurofilament heavy polypeptide, apolipoprotein D, and leucine-rich repeat-containing protein 4 expression were increased, most of which is consistent with the result of the transcriptomic analysis. Barrier-to-autointegration factor is a small DNA-binding protein [41] that plays a role in repairing DNA double-strand breaks [42] and nuclear ruptures [43], and its mutation causes premature aging syndrome [44]. Calcium/calmodulin-dependent protein kinase type II is the most abundant protein in excitatory synapses and is central to synaptic plasticity, learning, and memory [45]. Neurofilaments play an important role in axon radial growth and nerve conduction [46]; moreover, PACAP has been reported to stimulate the differentiation of size-sieved stem cells into neurons by increasing the expression of the neurofilament light polypeptide [47] and induce the differentiation of serum-cultured SH-SY5Y cells into neuroblastic cells by increasing the mRNA levels of three neurofilament proteins [48]. Furthermore, Apolipoprotein D is up-regulated at the mRNA and protein levels after treatment with PACAP [49]. Leucine-rich repeat-containing protein 4 is expressed in the hippocampus, promotes neurite extension in hippocampal neurons, and is involved in neuronal and glial cell differentiation [50]. Proteome sequencing confirmed that these proteins, which have been validated to be associated with PACAP and play neuroprotective and nerve-regenerative roles in the nervous system, also exhibit increased expression following treatment with SPAM1. Since our previous report confirmed that SPAM1 can re-set the PACAP-PAC1R system by up-regulating the expression of both PACAP and PAC1-R [9], the results of our multi-omic study not only demonstrate the positive activities of SPAM1 in the nervous system, such as neuroprotection and nerve regeneration, but also help to confirm the positive regulating effect of SPAM1 on re-setting the PACAP-PAC1R system. Several corresponding signal pathways were also revealed by our multi-omic analysis, including the neurotrophin signaling pathway (which is consistent with our SPAM1 data in RGC-5 cells [9]), circadian entrainment, olfactory transduction, and GnRH signaling, which is regulated by PACAP [51].

Finally, using Western blot analysis, we confirmed that SPAM1 up-regulates the expression of neurofilament light polypeptide and neurofilament medium polypeptide, which was indicated in the proteome analysis; thus, our laboratory test confirms that treatment with SPAM1 can promote nerve regeneration. It should be mentioned that the result of the Western blot analysis showed that the effect of 0.1 µmol/kg/day SPAM1 was more significant than that of 100 µmol/kg/day SPAM1. In fact, the positive allosteric modulation site of PAC1-R may be able to detect some type of stress metabolites, such as PACAP (28–38), which is a natural metabolite of the stress hormone PACAP38. The working mechanism of SPAM1 involves the imitation (mimicry) of some type of stress metabolites, such as PACAP (28–38), to trigger self-defensive responses in the nervous system, including anti-senescence, neuroprotection, and nerve regeneration. Thus, treatment with SPAM1 may have a similar effect to physical exercise. While too much exercise may not benefit the body, exercising regularly is important. In this study, SPAM1 was used every day, and we found that a low dose (0.1 μmol/kg/d) may be enough, while 100 μmol/kg/d may be too much. Thus, in future research, the in vivo effect of SPAM1 at a dose of 100 μmol/kg/week will be tested and evaluated.

Our whole-transcriptome and proteome sequencing results confirmed that SPAM1 exerts its biological functions through known signaling pathways, such as the cAMP signaling pathway and calmodulin pathway, and suggested that it exerts its neuroprotective and anti-aging effects through other pathways; this information will be helpful in further explorations of the mechanism of action of SPAM1.

## 4. Materials and Methods

### 4.1. Materials and Cell Lines

Mouse retinal ganglion cell (RGC-5) lines were provided by the Chinese Academy of Life Sciences (Shanghai, China). Peptide SPAM1 was synthesized by GL Biochem Ltd. (Shanghai, China) at 95% purity. The purity of the peptides was confirmed using reversed-phase high-performance liquid chromatography (HPLC), and they were characterized using matrix-assisted laser desorption/ionization time-of-flight (MALDI-TOF) mass spectrometry.

### 4.2. Grouping and Treatment of Animals

Male C57BL/6 mice (6 weeks old), purchased from Guangzhou Red Carrot Biotechnology Co., Ltd. (Guangzhou, China), were allowed to eat and drink freely in a clean environment with 12 h of light at 24 °C ± 1 °C and 55 ± 5% humidity. The mice were randomly divided into 6 groups. After 10 days of acclimatization to their new environment, the mice were continuously medicated for 6 weeks with two injections per day. The first was an injection of D-galactose (D-gal) (150 mg/kg/day) or saline, and SPAM1 or saline was injected 5 min after the first injection. Both injections were intraperitoneal; the first injection was in the left peritoneal cavity and the second in the right. The mice were grouped as follows: (1) normal control group (NOR): saline without D-gal; (2) D-gal group: D-gal + saline; (3) 0.1 µmol/kg/day SMAP1 group: D-gal + 0.1 µmol/kg/day SMAP1; (4) 1 µmol/kg/day SMAP1 group: D-gal + 1 µmol/kg/day SMAP1; (5) 10 µmol/kg/day SMAP1 group: D-gal + 1 µmol/kg/day SMAP1; (6) 100 µmol/kg /day SMAP1 group: D-gal + 100 µmol/kg/day SMAP1.

### 4.3. Tissue Preparation

After the completion of dosing on day 42, the mice were euthanized and weighed, and their brains were quickly removed and washed with RNase-free water. Some brain tissue was placed in 4% paraformaldehyde solution, followed by paraffin embedding, and the remaining tissue was immediately added to liquid nitrogen and subsequently transferred to −80 °C for storage.

### 4.4. Hematoxylin–Eosin (HE) Staining

The paraffin sections were placed in dewaxing solution and different concentrations of ethanol, stained with hematoxylin stain for 3–5 min, and stained with eosin for 5 min. Ethanol and xylene were used for dehydration, and the sections were sealed with neutral gum.

### 4.5. Senescence-Associated β-Galactosidase Staining

The RGC-5 cells were plated and cultured for a set duration in six-well plates. When the cells reached a confluence rate of 80%, the experimental groups were subjected to treatment with SPAM1 (1 µM–100 µM) for 24 h. The same volume of fetal bovine serum-free basal medium was administered to the control group without any treatment. The cell culture medium was aspirated, washed once with PBS, and then fixed and stained according to the instructions of the cell senescence β-galactosidase staining kit (Beyotime, Shanghai, China). After incubation overnight at 37 °C, the percentage of stained senescent cells was obtained by observing and counting under an ordinary light microscope. All experiments were performed with at least three parallel replicates and repeated three times.

### 4.6. Western Blotting Assays

RGC-5 cells in the logarithmic growth phase were inoculated into 6-well culture plates with 2 × 10^5^ cells per well and cultured in F12 medium containing 10% fetal bovine serum at 37 °C, 5% CO_2_, and 80% fusion. The experimental group was treated with SPAM1 (1–100 µM) for 24 h. The control group was treated with fetal bovine serum-free basal medium. Total protein was extracted from the cells using RIPA buffer (50 mM Tris-HCl (pH 7.4), 150 mM NaCl, 20 mM EDTA, 1% Triton X-100, 1% sodium deoxycholate,1% SDS, and protease inhibitors (Beyotime, Shanghai, China), placed on ice for 30 min, and analyzed using sodium dodecyl sulfate-polyacrylamide gel electrophoresis. The membranes were co-incubated with the following polyclonal antibodies: anti-SIRT6 antibody (12486S; Cell Signaling Technology, Danvers, MA, USA), anti-YY1 antibody (sc-7341; Santa Cruz Biotechnology, Shanghai, China), Lamin B1 Polyclonal antibody (12987-1-AP; Proteintech, Wuhan, China), anti-P16 antibody (AB51243; abcam, Shanghai, China), anti-NEFL antibody (2837T; Cell Signaling Technology, Beyotime, MA, USA), anti-NEFM antibody (67255S; Cell Signaling Technology, Beyotime, MA, USA), and anti-β-actin antibody (EM21002, HUABIO, Hangzhou, China). They were then incubated with HRP-coupled secondary antibody (HA1006, HUABIO, Hangzhou, China; ab150075, abcam, Shanghai, China). The protein bands were displayed using an enhanced chemiluminescence (ECL) kit (Beyotime Biotech, Shanghai, China). All experiments were performed with at least three parallel replicates and repeated three times.

### 4.7. Immunohistochemistry

The paraffin sections were added to dewaxing solution and different concentrations of ethanol. Antigen-repairing solution was used for antigen repairing, serum or BSA was used for sealing, SIRT6 was detected using an SIRT6 polyclonal antibody (13572-1-AP; Proteintech), YY1 was detected using an anti-YY1 antibody (sc-7341; Santa Cruz Biotechnology), Lamin B1 was detected using a Lamin B1 polyclonal antibody (12987-1-AP; Proteintech), and P16 was detected using an anti-P16 antibody (sc-1661; Santa Cruz Biotechnology). Then, the corresponding secondary antibody was added; the color was developed using diaminobenzidine; the nuclei of the cells were restained using hematoxylin; the cells were dehydrated with ethanol, n-butyl alcohol, and xylene; and the slices were sealed using sealer adhesive.

### 4.8. Whole-Transcriptome Sequencing

In total, 3 samples were randomly selected from the D-gal group (subsequently named the DA group), and 3 were randomly selected from the 100 µmol/kg/day SMAP1 group (subsequently named the SP group). Six mouse brain tissue samples were subjected to total RNA extraction by Genepioneer Biotechnologies Ltd. (Nanjing, China) and sequenced using an Illumina Nova 6000 platform and a NovaSeq 2500 platform after the libraries were tested and approved. The raw sequencing data were filtered with fastp (https://github.com/OpenGene/fastp) (accessed on 28 August 2023) and Cutadapt (v2.10) [52] to obtain high-quality clean data. The clean reads were mapped to the Mus musculus genome (http://ftp.ensembl.org/pub/current_fasta/mus_musculus/) (accessed on 29 August 2023) using HISAT2 (v2.1.0) [53] and Bowtie (v1.2.2) (http://bowtie-bio.sourceforge.net/index.shtml) (accessed on 29 August 2023) (mRNAs, lncRNAs and circRNAs were aligned to the reference genome using HISAT2 software. miRNAs were aligned to the reference genome using Bowtie (v1.2.2)). For the analysis of differentially expressed RNA, we used the DESeq2 software (R script) (1.26.0) to calculate differential folds. Different RNAs were screened for differences by selecting the appropriate metrics according to their specificity (see Section 2.4). We used R’s (V3.6.2) ggplot2 package to draw volcano maps and the (V3.6.2) pheatmap package to draw clustering heatmaps of differentially expressed mRNAs, lncRNAs, and circRNAs. The Python program developed by Genepioneer Biotechnologies Ltd. (Nanjing, China) was used to draw a heatmap of differentially expressed miRNAs’ clustering. Differentially expressed RNAs were functionally annotated and enriched using the Gene Ontology (GO) (http://www.geeontology.org) (accessed on 29 August 2023) and KEGG (Kyoto Encyclopedia of Genes and Genomes) (http://www.kegg.jp) (accessed on 29 August 2023) databases. For differentially expressed mRNAs, lncRNAs, and circRNAs, differentially expressed RNAs annotated to the KEGG and GO databases were enriched using Perl scripts from Genepioneer Biotechnologies Ltd. (Nanjing, China). The KEGG and GO enrichment analyses of differentially expressed miRNA target genes were performed according to the principle of hypergeometric distribution, and the Perl program was used to draw the statistical plots of the GO annotations of differentially expressed miRNA target genes. Additionally, KEGG enrichment scatter plots of differentially expressed miRNA target genes were drawn using the ggplot2 package in R (v3.6.2).

### 4.9. Proteomic Analysis

Three samples each from the D-gal group (subsequently named the DA group) and 100 µmol/kg/day SMAP1 group (subsequently named the SP group) were randomly selected. Six mouse brain tissue samples were handed over to Genepioneer Biotechnologies Ltd. (Nanjing, China) for protein extraction and quantification, and a TMT proteome assay was completed. The Mus musculus database (http://ftp.ensembl.org/pub/release-106/fasta/mus_musculus/) (accessed on 9 August 2023) was searched for mass spectrometry downstream data using the Proteome Discovery software (version: 3.0), and spectral peptides and proteins were quantified. Protein functional annotation was performed using the BLAST software (version: 2.2.26) (https://blast.ncbi.nlm.nih.gov/Blast.cgi) (accessed on 9 August 2023). For differential protein analysis, we used the Perl program developed by Genepioneer Biotechnologies Ltd. (Nanjing, China) to screen differentially expressed proteins based on multivariate and univariate statistical analysis. We used R’s (V3.6.2) ggplot2 package to draw volcano maps and the (V3.6.2) pheatmap package to draw clustering heatmaps of differentially expressed proteins. An interaction analysis of the identified proteins was performed using the StringDB protein interaction database (http://string-db.org/) (accessed on 9 August 2023), and the Gene Ontology (GO) (http://www.geeontology.org) (accessed on 9 August 2023) and KEGG (Kyoto Encyclopedia of Genes and Genomes) databases (http://www.kegg.jp) (accessed on 9 August 2023) were used for the functional annotation and enrichment analysis of differentially expressed proteins. For the differentially expressed proteins, the Perl program from Genepioneer Biotechnologies Ltd. (Nanjing, China) was used to annotate the differentially expressed proteins according to all the protein annotation results and to draw the pathway maps.

### 4.10. Statistical Analyses

GraphPad Prism 8 was used for statistical analysis. All data are expressed as mean ± standard error of the mean (SEM). One-way analysis of variance (ANOVA) was used to assess the differences between groups. When *p* < 0.05, the difference was statistically significant.

## 5. Conclusions

The present study confirmed that SPAM1 is able to restore the structure and neuron number of the hippocampus in a D-gal-induced aging mouse model. In vitro and in vivo tests showed that SPAM1 resists the senescence of neurons by up-regulating the expression of SIRT6 and Lamin B1 and down-regulating the expression of YY1 and p16.

In this study, we combined whole-transcriptome analysis and proteome analysis to further investigate the mechanism of SPAM1’s action. The results of these analyses suggest that SPAM1 acts through the neuroactive ligand–receptor interaction pathway, longevity-regulating pathway—multiple species, longevity-regulating pathway, cell cycle pathway, etc.

It was demonstrated that SPAM1 can re-set the PACAP-PAC1R system by up-regulating PACAP and PAC-1R expression [9], and the results of the whole-transcriptome and proteome analyses indicate that SPAM1 is indeed associated with the PACAP-PAC1R-related pathways; for example, SPAM1 works through the cAMP signaling pathway, calcium signaling pathway, and neurotrophin signaling pathway.

Except the conventional pathways mediated by the PACAP-PAC1R system, it was confirmed for the first time that SPAM1 induces some pathways associated with anti-aging, such as the longevity-regulating pathways; this may be attributed to the positive allosteric modulation role of SPAM1 in PAC1-R, whereby it imitates the stress metabolite to trigger self-defensive responses in the nervous system, including anti-senescence, neuroprotection, and nerve regeneration.

Finally, the Western blot analysis confirmed that SPAM1 up-regulates the expression of neurofilament light polypeptide and neurofilament medium polypeptide, which supports the proteomics results.

The results of this study lay a pharmacological network foundation for the use of SPAM1 as a potent therapeutic anti-aging drug against neurodegeneration with anti-senescence, neuroprotective, and nerve regeneration activity.

## Figures and Tables

**Figure 1 ijms-25-03872-f001:**
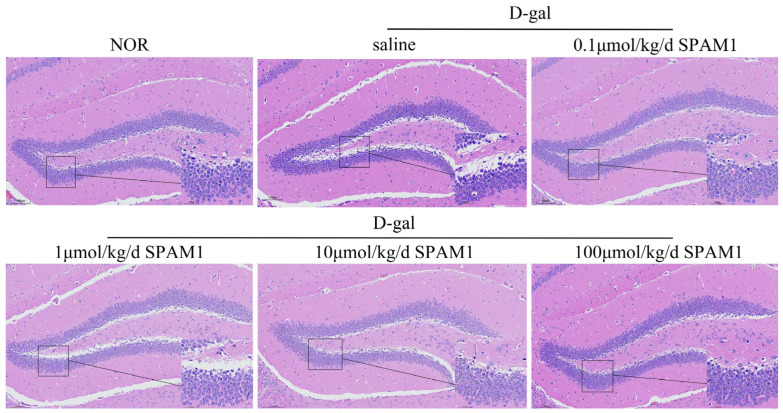
SPAM1 attenuates the D-gal-induced reduction in the number of mouse hippocampal neurons. HE staining showed that D-gal treatment led to a reduction in mouse hippocampal neurons, which was most significantly reversed by SPAM1 at concentrations of 0.1 μmol/kg/day and 100 μmol/kg/day.

**Figure 2 ijms-25-03872-f002:**
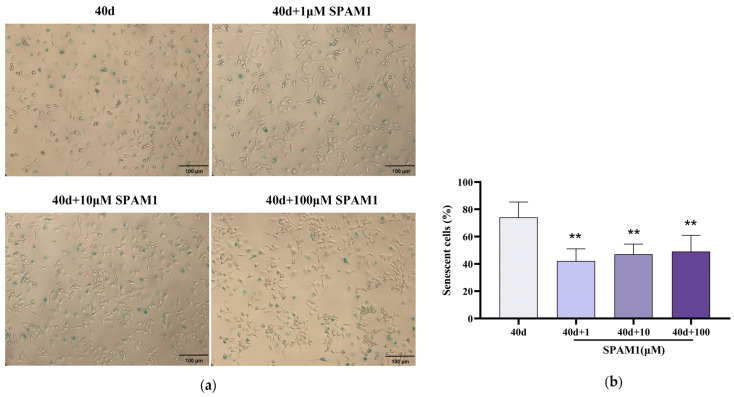
SPAM1 ameliorated RGC cell senescence. (**a**) β-gal staining and (**b**) corresponding statistics showed that the number of β-gal-positive RGC-5 cells was significantly reduced by different concentrations of SPAM1. ** *p* < 0.01 vs. 40 d. Data are presented as the mean ± SEM of three experiments.

**Figure 3 ijms-25-03872-f003:**
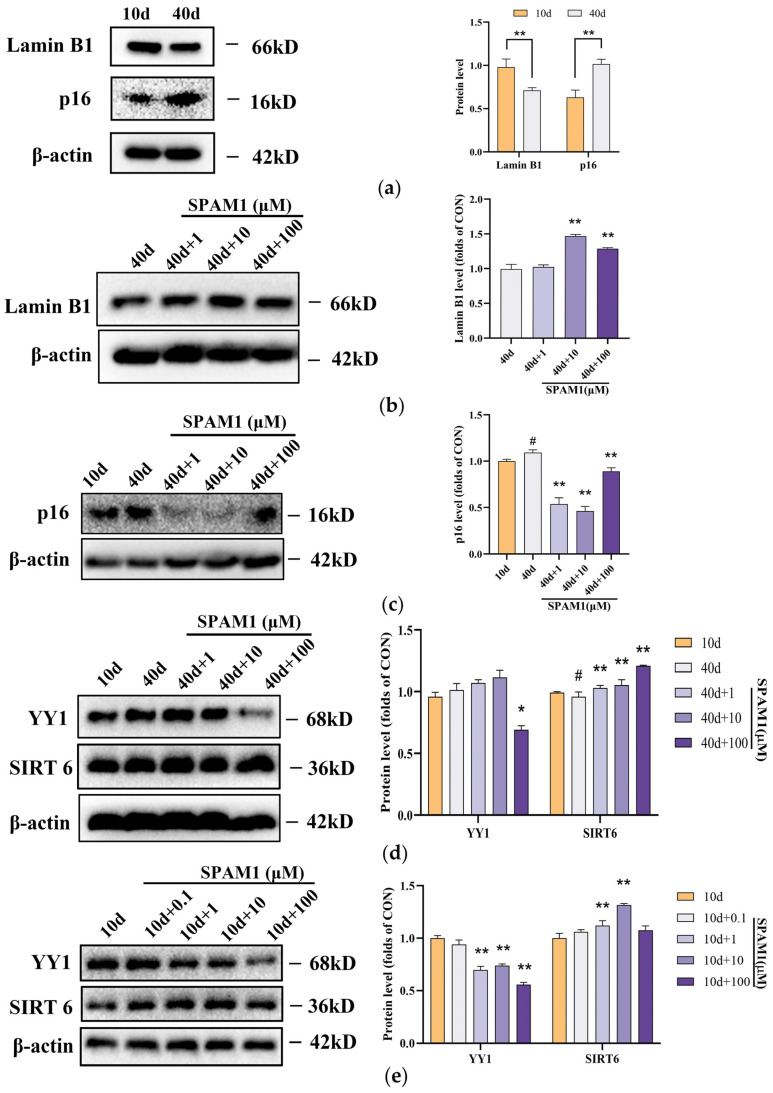
Effect of SPAM1 on the expression of SIRT6, YY1, Lamin B1, and p16 in RGC-5 cells. (**a**) WB images of Lamin B1 and p16 in RGC-5 whole cells (**left**) and the corresponding statistics (**right**) show that the expression of Lamin B1 was reduced and p16 expression was increased in RGC-5 cells after 40 days. ** *p* < 0.01 vs. 10 d. (**b**) WB images (**left**) and corresponding statistics (**right**) of Lamin B1 in RGC-5 whole cells after 40 days show that a 10~100 μM concentration of SPAM1 significantly increased the expression of Lamin B1. ** *p* < 0.01 vs. 40 d. (**c**) WB images (**left**) and corresponding statistics (**right**) of p16 in RGC-5 whole cells show that different concentrations of SPAM1 could down-regulate the expression of p16. # *p* < 0.05 vs. 10 d; ** *p* < 0.01 vs. 40 d. (**d**) WB images (**left**) and corresponding statistics (**right**) of YY1 and SIRT6 in RGC-5 whole cells show that a 100 μM concentration of SPAM1 decreased the expression of YY1 and 1 to 100 μM of SPAM1 up-regulated the expression of SIRT6. # *p* < 0.05 vs. 10 d; * *p* < 0.05 vs. 40 d; ** *p* < 0.01 vs. 40 d. (**e**) WB images (**left**) and corresponding statistics (**right**) of YY1 and SIRT6 in RGC-5 whole cells (10 days) show that a 100 μM concentration of SPAM1 decreased the expression of YY1. ** *p* < 0.01 vs. 10 d. Data are presented as the mean ± SEM of three experiments.

**Figure 4 ijms-25-03872-f004:**
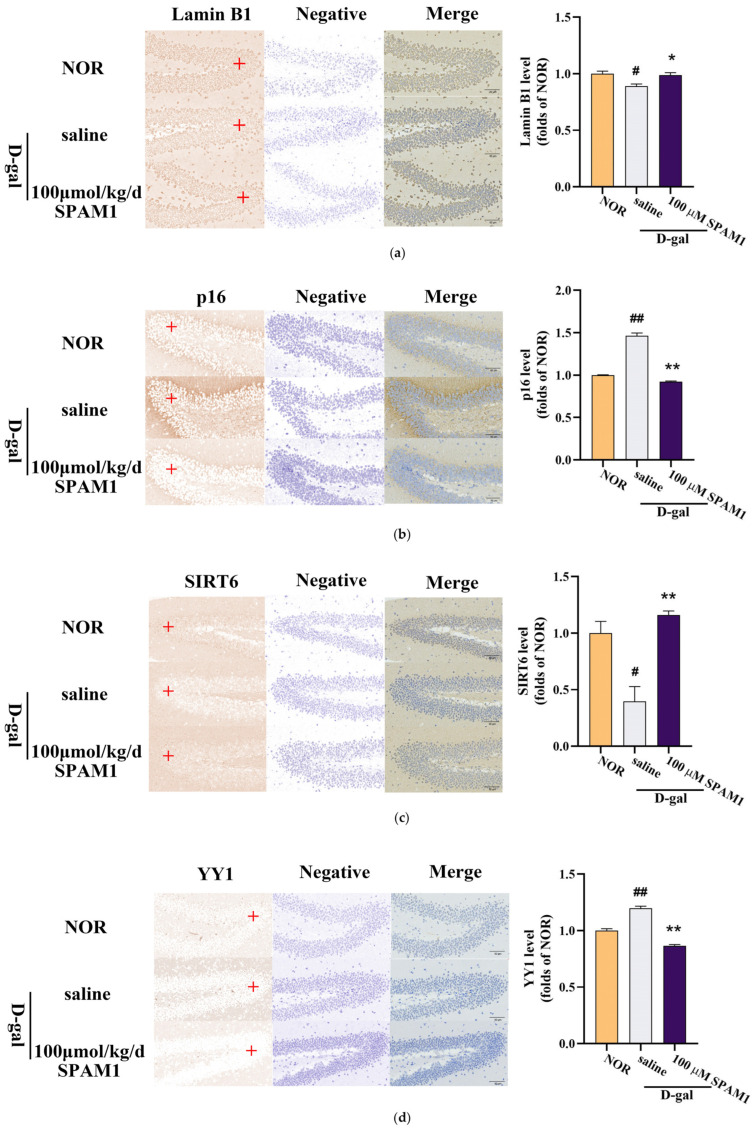
Effect of SPAM1 treatment on the expression of SIRT6. (**a**) Immunohistochemical images of Lamin B1 expression in the hippocampus (**left**) and corresponding statistical chart (**right**) of Lamin B1 expression in the hippocampus. (**b**) Immunohistochemical images of p16 expression in the hippocampus (**left**) and corresponding statistical chart (**right**) of p16 expression in the hippocampus. (**c**) Immunohistochemical images of SIRT6 expression in the hippocampus (**left**) and corresponding statistical chart (**right**) of p16 expression in the hippocampus. (**d**) Immunohistochemical images of YY1 expression in the hippocampus (**left**) and corresponding statistical chart (**right**) of YY1 expression in the hippocampus. The data for the statistical graph were taken from the red + areas in the graph above. # *p* < 0.05 vs. NOR; ## *p* < 0.01 vs. NOR; * *p* < 0.05 vs. saline; ** *p* < 0.01 vs. saline. Data are presented as the mean ± SE, *n* = 8–10.

**Figure 5 ijms-25-03872-f005:**
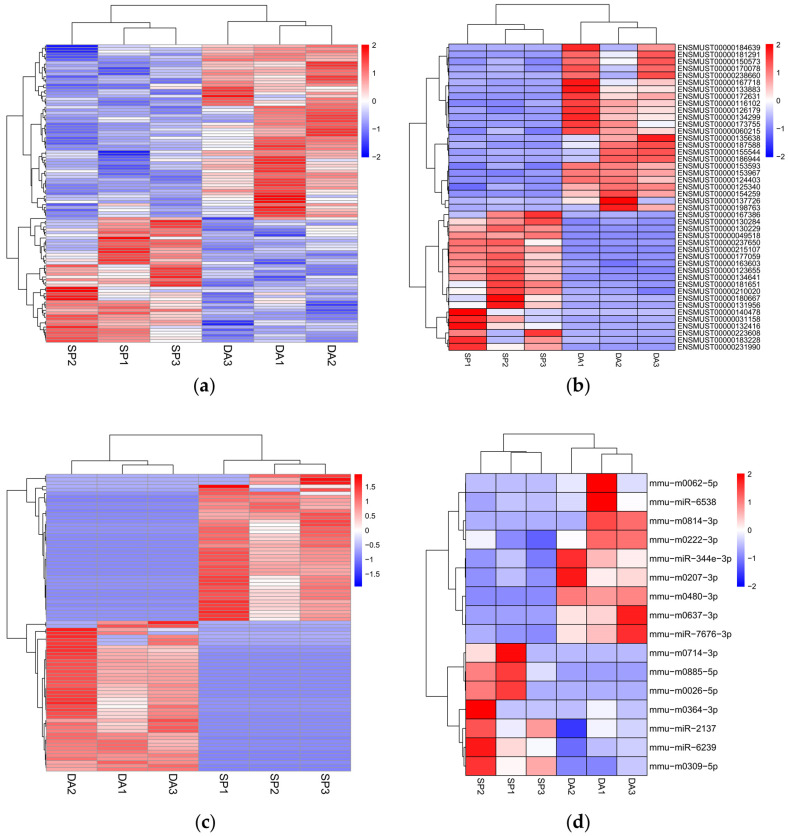
Heatmaps of differentially expressed mRNAs, lncRNAs, circRNAs, and miRNAs in DA and SP groups. (**a**) Heatmap of differentially expressed mRNAs; (**b**) heatmap of differentially expressed lncRNAs; (**c**) heatmap of differentially expressed circRNAs; (**d**) heatmap of differentially expressed miRNAs.

**Figure 6 ijms-25-03872-f006:**
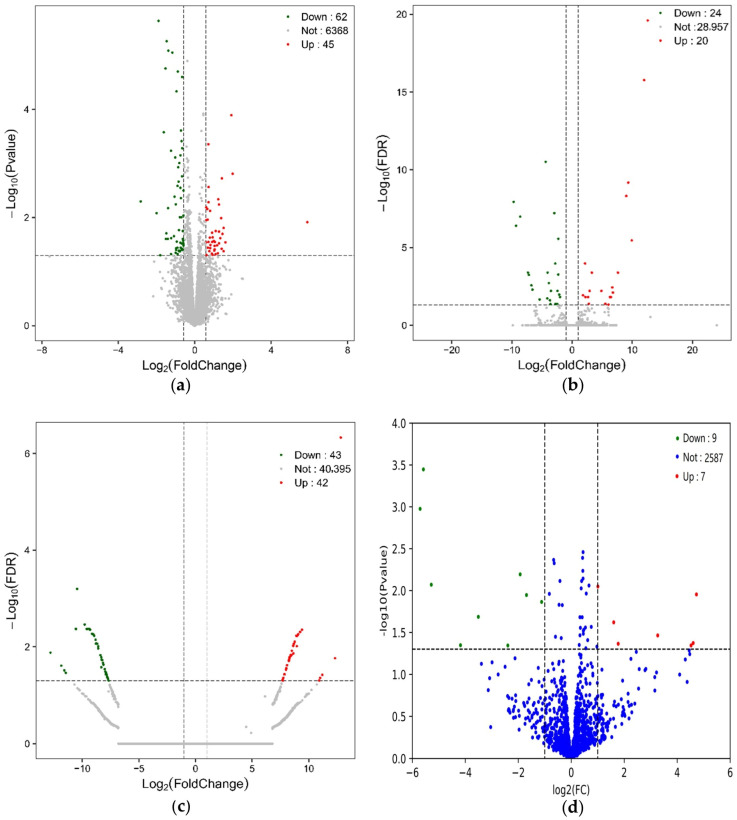
Volcano plots of differentially expressed mRNAs, lncRNAs, circRNAs, and miRNAs in DA and SP groups. (**a**) Volcano plot of 107 differentially expressed mRNAs; (**b**) volcano plot of 44 differentially expressed lncRNAs; (**c**) volcano plot of 85 differentially expressed circRNAs; (**d**) volcano plot of 16 differentially expressed miRNAs.

**Figure 7 ijms-25-03872-f007:**
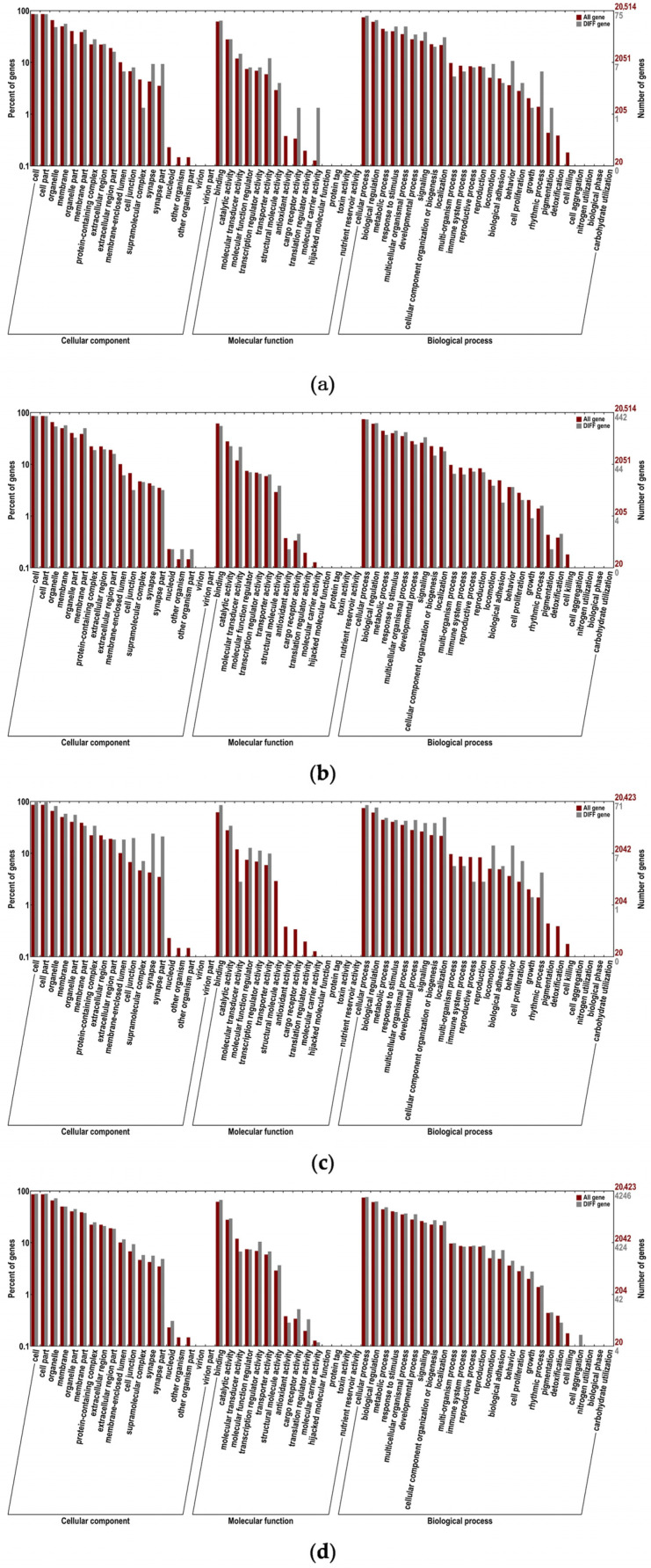
GO enrichment analysis of differentially expressed mRNAs, differentially expressed LncRNA target genes, host genes that differentially express circRNAs, and differentially expressed miRNA target genes. GO biological functional analyses of (**a**) differentially expressed mRNAs; (**b**) differentially expressed LncRNA target genes; (**c**) host genes that differentially express circRNAs; (**d**) differentially expressed miRNA target genes.

**Figure 8 ijms-25-03872-f008:**
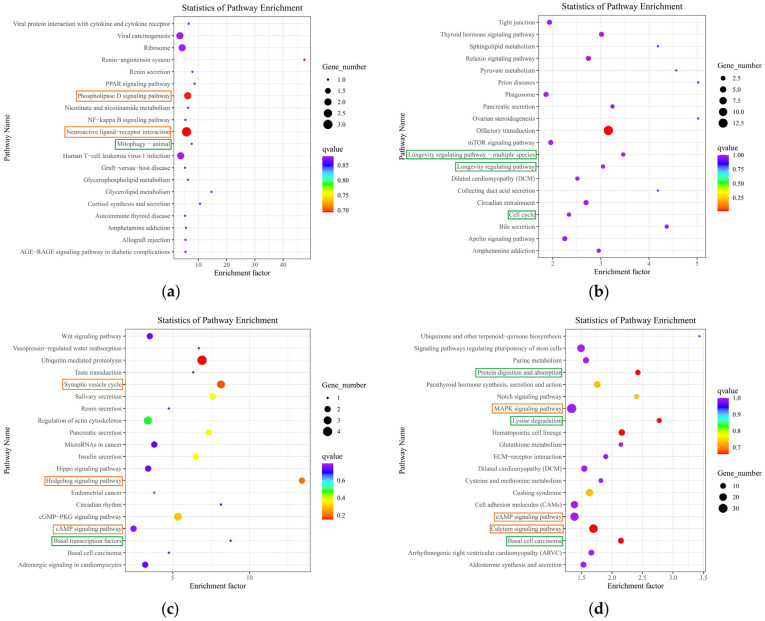
KEGG pathway enrichment analysis of differentially expressed mRNAs, differentially expressed LncRNA target genes, host genes that differentially express circRNAs, and differentially expressed miRNA target genes. KEGG pathway analyses of (**a**) differentially expressed mRNAs; (**b**) differentially expressed LncRNA target genes; (**c**) host genes that differentially express circRNAs; (**d**) differentially expressed miRNA target genes. The signaling pathways in the orange boxes represent conventional pathways activated by PACAP, and those in the green boxes are strongly associated with anti-aging and longevity, but are not involved in the conventional PACAP pathways.

**Figure 9 ijms-25-03872-f009:**
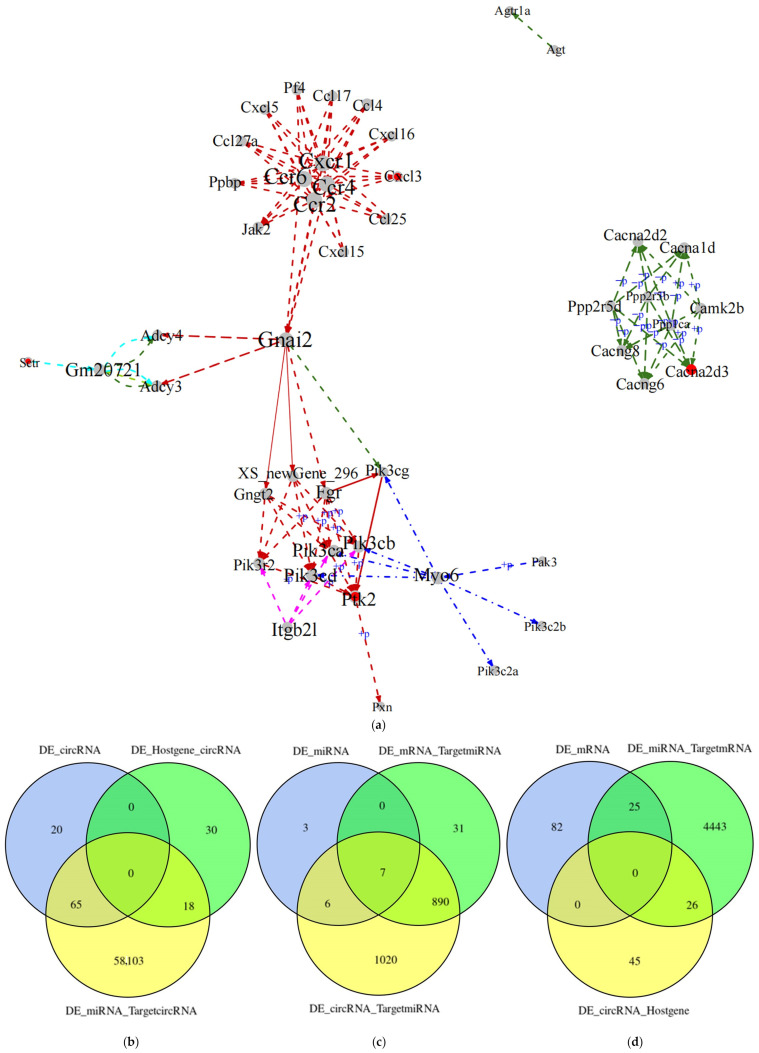
Whole-transcriptome association analysis. (**a**) KEGG integrated pathway network. Each dot represents a gene, and lines represent the relationships among genes or between genes and pathways. The colors of the different lines represent relationships between genes and different pathways. The red dots are key genes. Combined analysis of differentially expressed RNA targeting relationships. (**b**) Analysis of host genes that differentially express circRNAs and their associated target RNAs; (**c**) analysis of target RNAs associated with differentially expressed miRNAs; (**d**) analysis of target RNAs associated with differentially expressed mRNAs. DE_circRNA: differentially expressed circRNAs; DE_Hostgene_circRNA: all circRNAs with differentially expressed genes as the host gene; DE_miRNA_TargetcircRNA: all circRNAs targeted by differentially expressed miRNAs; DE_miRNA: differentially expressed miRNAs; DE _mRNA_TargetmiRNA: all miRNAs targeted by differentially expressed mRNAs; DE_circRNA_TargetmiRNA: all miRNAs targeted by differentially expressed circRNAs; DE_mRNA: differentially expressed mRNAs; DE_miRNA_TargetmRNA: all target genes of differentially expressed miRNAs; DE_circRNA_Hostgene: differential expression of circRNAs for all host genes.

**Figure 10 ijms-25-03872-f010:**
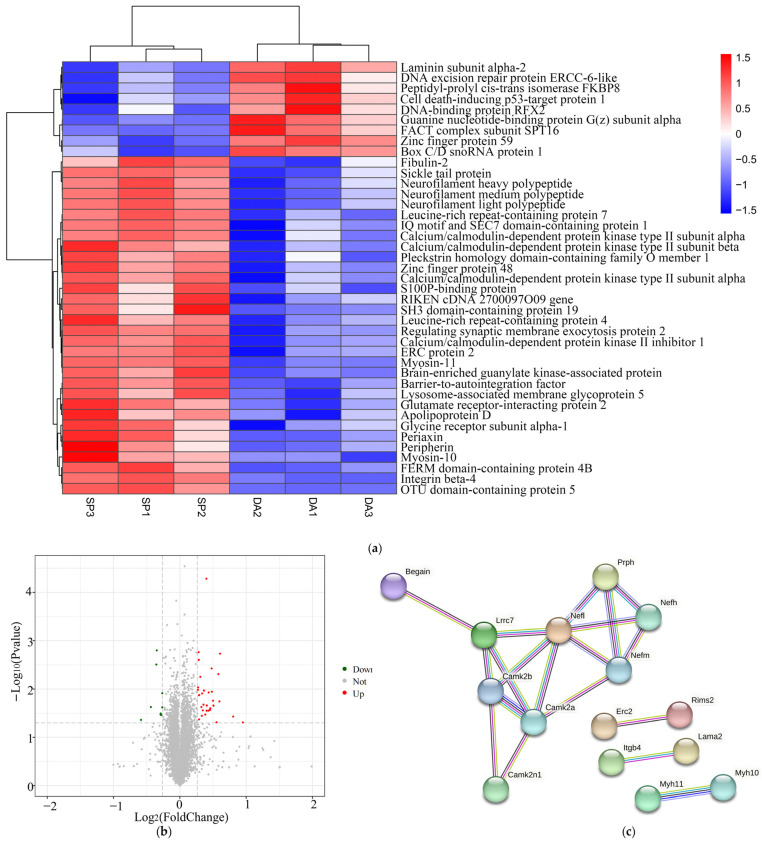
Heatmaps and volcano plots of differentially expressed proteins in DA and SP groups. (**a**) Heatmap of differentially expressed proteins; (**b**) volcano plot of differentially expressed proteins. (**c**) Differentially expressed protein string network graph. Each node represents a protein, with thicker lines representing higher association confidence.

**Figure 11 ijms-25-03872-f011:**
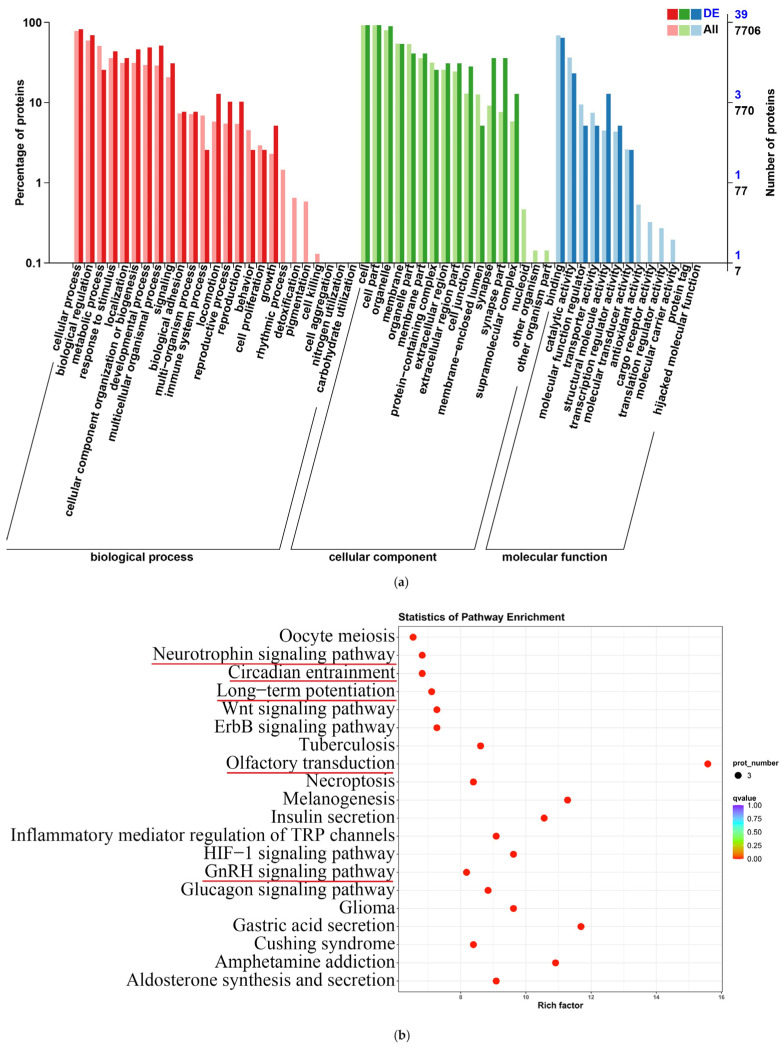
GO and KEGG pathway enrichment analyses of differentially expressed proteins. (**a**) GO biological functional analyses of differentially expressed proteins; (**b**) KEGG pathway analyses of differentially expressed proteins. Signaling pathways marked with a red bar indicate an association with longevity, neuroprotection, or the PACAP-PAC1 signaling pathway.

**Figure 12 ijms-25-03872-f012:**
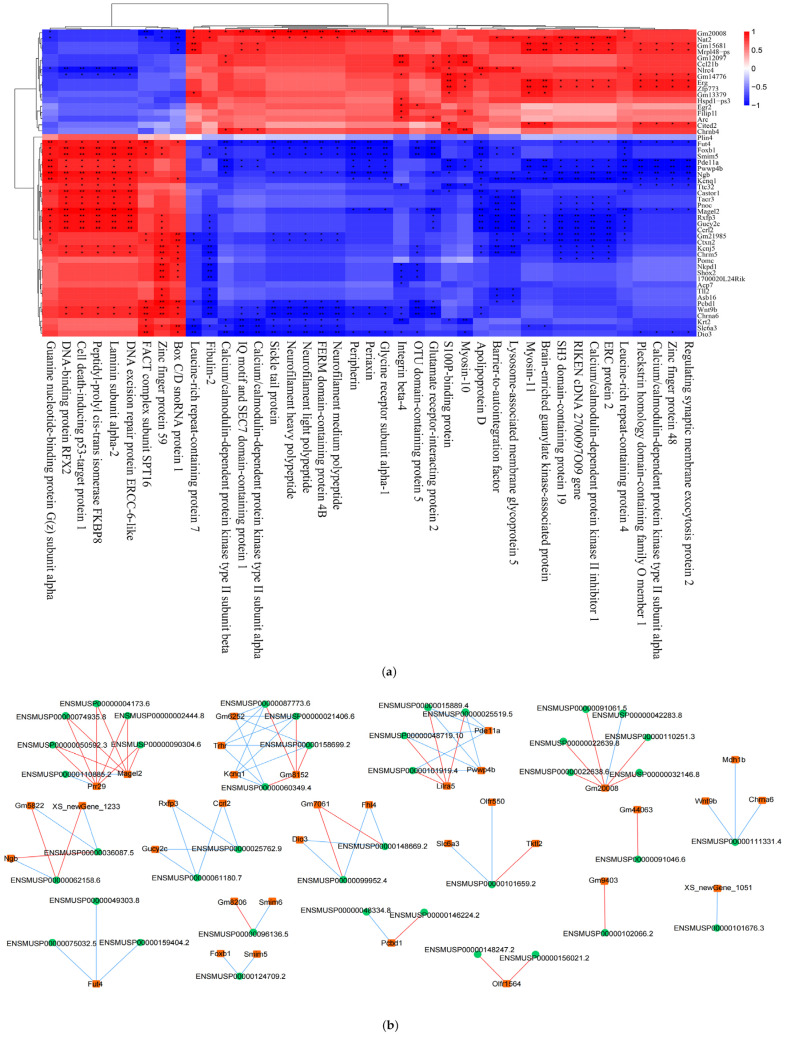
(**a**) Spearman correlation heatmap. The columns represent differential proteins, the rows represent differential genes, and the magnitude of the correlation is represented by the color. * *p* < 0.05; ** *p* < 0.01. (**b**) Network diagram. The rectangles in the diagram are differential genes, the circles are differential proteins, the blue lines indicate negative correlations, and the red lines indicate positive correlations.

**Figure 13 ijms-25-03872-f013:**
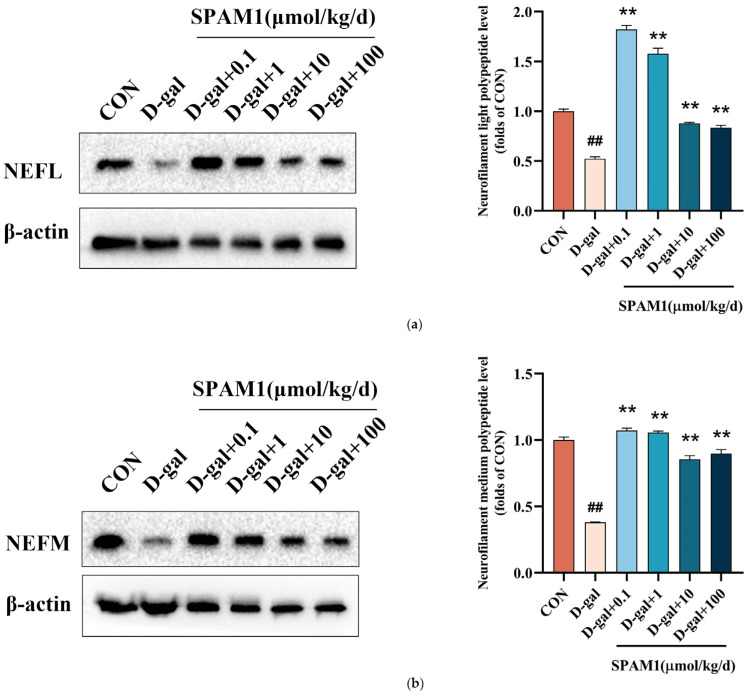
Validation of differentially expressed proteins. (**a**) WB images (**left**) and corresponding statistics (**right**) of neurofilament light polypeptide (NEFL) in D-gal model mouse brain tissue show significant up-regulation of neurofilament light polypeptide (NEFL) expression at different concentrations of SPAM1. (**b**) WB images (**left**) and corresponding statistics (**right**) of neurofilament medium polypeptide (NEFM) in D-gal model mouse brain tissue show significant up-regulation of neurofilament medium polypeptide (NEFM) expression at different concentrations of SPAM1. ## *p* < 0.01 vs. CON; ** *p* < 0.01 vs. D-gal. Data are presented as the mean ± SEM of three experiments.

## Data Availability

Data are contained within the article.

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
