# Peer review of "Integrated Transcriptomic and Proteomic Study of the Mechanism of Action of the Novel Small-Molecule Positive Allosteric Modulator 1 in Targeting PAC1-R for the Treatment of D-Gal-Induced Aging Mice"

_ijms, 2024, doi:10.3390/ijms25073872_

Round 1

Reviewer 1 Report

Comments and Suggestions for Authors

The Pituitary adenylate cyclase-activating polypeptide receptor 1 (PAC1-R) is a G protein-coupled receptor (GPCR) known for its affinity towards pituitary adenylate cyclase-activating polypeptide (PACAP) and its role in various neurological processes. PAC1-R is highly expressed in neurogenic regions of the brain and its activation mediates anti-apoptotic, antioxidant, and neurogenic effects, showing promise in mitigating neurodegenerative diseases like Alzheimer's and Parkinson's. Previous research identified SPAM1 as high affinity agonist for PAC1-R, exhibiting neuroprotective effects.

In this study, authors find SPAM1 to up-regulate SIRT6 and Lamin B1 expression while down-regulating YY1 and p16 expression in the hippocampus of an aging mouse model induced by D-galactose. Whole transcriptome sequencing and proteomic analysis further elucidated SPAM1's neuroprotective mechanism, laying the groundwork for its potential therapeutic application in neurodegenerative diseases.

The study gives provides comprehensive high quality data on the action of SPAM1 and I recommend publication in IJMS if authors address the minor concerns below:

1)        Please could authors rewrite the Abstract. It is not to up to standard of a scientific publication.

2)        It is unusual to have the lettering for different panels underneath the panels. Usually they are placed in the left upper corner of the panel.

3)        While a quantification is described in the legend to figure 1, there is not such panel in the figure.

4)        Figure legend 4 is not corresponding to what is in the figure. SMAP1 should read SPAM1. SIRT6 is only in panel c, not in a,b and d as stated in figure legend.

5)        Line 155: the shortenings DA and SP are not explained.

6)        Line 238: The meaning of this sentence is not clear to me. What are “genes in key RNAs”? Could authors please reformulate the statement made in this sentence.

7)        Line 241-256: Could authors please rewrite these lines. The statements seem non-sensical to me, apart from that they are grammatically not full sentences.

8)        Line 275: is the fold change actual fold change or log2 (as in figures)? Why is 0.83 used as corresponding to 1.2 and not 0.8?

9)        Line 379: the first eight words are a repetition of the last eight words in line 378.

10)  Line 379-383: Please rewrite sentence (it is very difficult to understand in the wording chosen by authors), in particular the half-sentence after ‘but’ in lines 382-383.

11)  Line 402: the three words ‘were enriched in’ are repeated. Please remove.

12)  Line 416-419: These two sentences, while in itself correct, are not integrated into the overall discussion of aging, neither is sentence in lines 431-432. Maybe it would be advantageous to put sentence in lines 432-437 first as argument, and then use information present in lines 416-419 and 431-432 to discuss why it is of interest that these particular molecules are affected by SPAM1.

13)  Figure 6: it is not clear why in two graphs is chosen an FDR<0.05 and in the other two a p<0.05 as threshold. Usage of FDR vs p value cannot be randomly selected. Authors should use only one parameter for all four plots or justify clearly why they need to use FDR vs p value in the respective graphs.

14)  Figure 8a-d and Figure 11b: it is not clear by which criteria some pathways are highlighted (boxes in Fig 8a-d, underlined in Fig 11b).

15)  Figure 9: line 259 refers to rectangles in Figure 9. I cannot see any. The Figure would be easier to understand if lines would be solid, instead of hashed.

16)  Figure 11a: Please could authors explain how right y-axis for DE is twice labelled “1”.

17)  Figure 13a: Could authors please comment on why 0.1 mmol/kg/d SPAM1 is more effective than 100 mmol/kg/d SPAM1.

Comments on the Quality of English Language

Some of the sentence and the abstract are not easy to read and have clearly not been cross-checked by someone very proficient in the English language.

Reviewer 2 Report

Comments and Suggestions for Authors

1. The description of the statistical analysis is not correct. For instance, how did the authors perform differential gene analysis? What packages and functions?

2. The same applies to the proteomic analysis, the identification of statistical differences, and the packages and functions used need to be described.

3. The pathway analysis is not described at all, how this was performed? 

4. Figures are complex, and while they are all relevant, the figures are not well-explained. The authors should explain every figure in greater detail. The figure legend should be sufficiently detailed to explain everything about the figure.

5. Please also describe the packages and functions that were used to generate the figures.

Comments on the Quality of English Language

The authors should check for typos and errors.

Round 2

Reviewer 1 Report

Comments and Suggestions for Authors

I am content with the implemented changes and recommend publication.

Author Response

感谢您的审阅和批准!

Reviewer 2 Report

Comments and Suggestions for Authors

The authors have significantly improved the manuscript, but I found one more thing:

What is this sentence: "The clean reads were compared to the Mus musculus genome (http://ftp.ensembl.org/pub/current_fasta/mus_musculus/) (accessed on 29 August 2023) using HISAT2 [53] and Bowtie (v1.2.2) (http://bowtie-bio.sourceforge.net/index.shtml) (accessed on 29 August 2023). "

Two questions:

1. The process is not called "comparison" but this is an alignment or mapping.

2. HISAT2 and Bowtie do exactly the same things, just in different ways. What did the authors use?
